# Atorvastatin and Fluvastatin Potentiate Blood Pressure Lowering Effect of Amlodipine through Vasorelaxant Phenomenon

**DOI:** 10.3390/medicina59061023

**Published:** 2023-05-25

**Authors:** Niaz Ali, Wajid Ali, Abid Ullah, Shujaat Ahmad, Ahad Amer Alsaiari, Mazen Almehmadi, Osama Abdulaziz, Mamdouh Allahyani, Abdulelah Aljuaid

**Affiliations:** 1Department of Pharmacology, College of Medicine, Shaqra University, Shaqra 11961, Saudi Arabia; 2Department of Pharmacology, Institute of Basic Medical Sciences, Khyber Medical University, Peshawar 25100, Khyber Pakhtunkhwa, Pakistan; 3Department of Pharmacy, Shaheed Benazir Bhutto University Sheringal, Dir Upper 18000, Khyber Pakhtunkhwa, Pakistan; abid@sbbu.edu.pk (A.U.); shujaat@sbbu.edu.pk (S.A.); 4Department of Clinical Laboratory, Sciences Saudi Arabia Department, College of Applied Medical Sciences, Taif University, Taif 21944, Saudi Arabia; ahadamer@tu.edu.sa (A.A.A.); mazenn@tu.edu.sa (M.A.); o.osama@tu.edu.sa (O.A.); m.allahyani@tu.edu.sa (M.A.); ab.aljuaid@tu.edu.sa (A.A.)

**Keywords:** statins, atorvastatin, fluvastatin, amlodipine, verapamil, calcium channel blocking activity

## Abstract

*Background and Objectives:* We have recently reported that stains have calcium channel blocking activity in isolated jejunal preparations. In this study, we examined the effects of atorvastatin and fluvastatin on blood vessels for a possible vasorelaxant effect. We also studied the possible additional vasorelaxant effect of atorvastatin and fluvastatin, in the presence of amlodipine, to quantify its effects on the systolic blood pressure of experimental animals. *Materials and Methods:* Atorvastatin and fluvastatin were tested in isolated rabbits’ aortic strip preparations using 80mM Potassium Chloride (KCl) induced contractions and 1 micro molar Norepinephrine (NE) induced contractions. A positive relaxing effect on 80 mM KCl induced contractions were further confirmed in the absence and presence of atorvastatin and fluvastatin by constructing calcium concentration response curves (CCRCs) while using verapamil as a standard calcium channel blocker. In another series of experiments, hypertension was induced in Wistar rats and different test concentrations of atorvastatin and fluvastatin were administered in their respective EC_50_ values to the test animals. A fall in their systolic blood pressure was noted using amlodipine as a standard vasorelaxant drug. *Results:* The results show that fluvastatin is more potent than amlodipine as it relaxed NE induced contractions where the amplitude reached 10% of its control in denuded aortae. Atorvastatin relaxed KCL induced contractions with an amplitude reaching 34.4% of control response as compared to the amlodipine response, i.e., 39.1%. A right shift in the EC_50_ (Log Ca++ M) of Calcium Concentration Response Curves (CCRCs) implies that statins have calcium channel blocking activity. A right shift in the EC_50_ of fluvastatin with relatively less EC_50_ value (−2.8 Log Ca++ M) in the presence of test concentration (1.2 × 10^−7^ M) of fluvastatin implies that fluvastatin is more potent than atorvastatin. The shift in EC_50_ resembles the shift of Verapamil, a standard calcium channel blocker (−1.41 Log Ca++ M). *Conclusions:* Atorvastatin and fluvastatin relax the aortic strip preparations predominantly through the inhibition of voltage gated calcium channels in high molar KCL induced contractions. These statins also inhibit the effects of NE induced contractions. The study also confirms that atorvastatin and fluvastatin potentiate blood pressure lowering effects in hypertensive rats.

## 1. Introduction

Cardiovascular diseases (CVDs) are associated with increased mortality and morbidity [1]. Globally, about 80% of deaths occur due to CVDs. In 2017, 17.8 million deaths occurred due to CVDs [2]. CVDs most commonly includes hypertension, dyslipidemia, angina, myocardial infarction, and stroke [3]. They are often treated with single or multiple drug therapies. Research and epidemiological data show that hypertension and dyslipidemias cannot be sufficiently controlled with monotherapy, despite the availability of well tolerated antilipidemics and antihypertensives. However, the literature shows some evidence that a combination of antihypertensives and lipid lowering agents provides cardiovascular benefits [4,5]. The World Health Organization (WHO) approved drugs for the treatment of CVDs such as Beta blockers, Calcium Channel Blockers, Angiotensin receptor blockers, Statins, and antiplatelet drugs [6]. Statins inhibit the β-Hydroxy β-methylglutaryl-CoA (HMG-CoA) enzyme and reduce lipid levels [7]. According to research data, statins show some effects that are independent from its lipid lowering action. These effects are called the pleiotropic effects of statins. Some of statins’ pleiotropic effects are improving endothelial function, enhancing the stability of atherosclerotic plaques, decreasing oxidative stress, decreasing inflammation, and inhibiting thrombogenic response [8,9]. We have recently reported the new aspect for inhibition of voltage gated calcium channels in gut tissues [10]. Studies have shown that statins up-regulate the calcium channels in vascular cell lines [11,12]. A study conducted on rabbits’ intestines states that statins have inhibitory effects on voltage gated calcium channels [10]. CVD patients often suffer from co-morbid conditions such as hypertension and dyslipidemias. These co-morbid conditions are commonly treated with drug combinations such as calcium channel blockers and statins [13,14]. In light of the statins’ pleiotropic effects, and our recent reports on statins for inhibition of voltage gated calcium channels in gut tissues, the current study aims to examine the drug–drug interaction of atorvastatin, fluvastatin, and amlodipine on blood vessels. The study also outlines its possible additional effects on the blood pressure lowering effect in Wistar rats.

## 2. Materials and Methods

### 2.1. Study Settings

This study was conducted at the Pharmacology department, Institute of Basic Medical Sciences, Khyber Medical University, Peshawar, KP, Pakistan and at Shaqra University, College of Medicine, Shaqra, KSA.

### 2.2. Raw Materials, Drugs and Solutions, and Standards

Analytical-grade chemicals were used for the experiments. Acetylcholine (ACh) [15], Norepinephrine (NE), and Potassium chloride (KCL) were purchased from BDH, Poole, England. Atorvastatin raw material was purchased from Polyfine Pharmaceutical Private Ltd., Peshawar, Pakistan. Fluvastatin of Novartis Pharma was purchased from the local market of Peshawar. Amlodipine raw material was obtained from Feroz-sons laboratories Pvt Ltd., Nowshera, Pakistan.

Less soluble raw materials in Krebs solution were suspended in 0.01% carboxy methyl cellulose (CM) using distilled water. To rule out any possible effects of CMC, a negative control in distilled water was conducted. Freshly prepared solutions and suspensions were used on the same day of the experiments.

### 2.3. Animals

Male and female rabbits weighing about 1.5–2.0 kg of a local breed were used for the experiments. The animals were housed in the animal house of Khyber Medical University. Free access to water was given. Overnight fasted rabbits were used on the day of the experiments. The Advanced Study and Research Board and Institutional Research Ethical Board approved this study protocol via Approval No. KMU/IBMS/IRBE/meeting/2022/9303-6 dated 12 October 2022. Moreover, Wistar rats were used for translating the in vitro vascular effects on systolic hypertension.

### 2.4. Data Recording

The effects on aortic strips were recorded using a Force Transducer (Model No: MLT 0225 Pan Lab S.1) attached via a bridge amplifier with 4 channels Power lab (Model No: 4/25T) purchased from AD Instruments, Australia. Lab chart 7 was used to record and interpret the vascular responses of the isolated aortic strip preparations. Systolic blood pressure was recorded via a tail cuff amplifier attached to the Power lab.

### 2.5. Physiological Solutions

Three types of Krebs solutions were used for the experiments: (1) Krebs normal solution consisting of ingredients in mM: NaCl 118.2, KCl 4.7, KH_2_PO_4_ 1.3, MgSO_4_ 1.2, NaHCO_3_ 25.0, Glucose 11.7, and CaCl_2_ 2.5. (2) Potassium Normal (Ca^++^ free) Krebs solution with ingredients in mM: NaCl 118.2, KCl 4.7, KH_2_PO_4_ 1.3, MgSO_4_ 1.2, NaHCO_3_ 25.0, Glucose 11.7, and ethylenediamine tetra acetic acid (EDTA) 0.1. (3) Potassium Rich (Ca^++^ free) with ingredients in mM: NaCl 50.58, KCl 50, KH_2_PO_4_ 1.26, MgSO_4_ 3.10, NaHCO_3_ 23.8, Glucose 11.1, and ethylenediamine tetra acetic acid (EDTA) 0.1.

Freshly prepared solutions in deionized water were used on the same day of the experiments.

### 2.6. Statins and Amlodipine Effects on 1 uM Norepinephrine (NE) Induced Contractions

Overnight fasted rabbits’ thoraxes were opened. Their aortae were removed and placed in Krebs solution contained in petri dishes aerated with a continuous supply of carbogen gas. Under a dissecting microscope, vessels were freed of adhering connective tissues and were sectioned into strips of 2 to 3 mm diameter using a sharp razor blade. Two types of aortic strips were prepared. Intact strips with intact endothelium and denuded strips, whose luminal surface was gently stroked with a moist cotton swab, subsequently confirmed by lack of ACh mediated relaxation. The strips were mounted in a tissue organ bath for the incubation period. Different concentrations of test atorvastatin, fluvastatin, and amlodipine were applied on the isolated tissue preparations. NE (1 μM) was added to the tissue organ baths of both intact and denuded aortic strips to maintain the tissue and obtain sustained contractions. Different test concentrations (10^−8^ to 10^−2^) M of atorvastatin, fluvastatin, and amlodipine were added in a cumulative manner. A gap of 1 min between test sample applications was provided. Changes in NE induced isometric tension were noted as per reported procedure [10,16,17]. The experiments were run four times. Mean Effective Concentrations (EC_50_) for statins and amlodipine were calculated.

### 2.7. Statins and Amlodipine Effects on KCl (80 mM) Induced Contractions

A relaxant effect on the isolated aortic strips sometimes involves inhibition of voltage gated calcium channels. Moreover, a relaxant effect on 80 mM KCl induced contractions are usually considered to follow relaxation via inhibition of voltage gated calcium channels [18,19,20]. Therefore, we produced sustained contractions in intact and denuded aortic strips via KCl (80 mM). A period of 40–60 min stabilization was allowed. Atorvastatin, fluvastatin, and amlodipine in similar test concentrations were applied in a cumulative manner in one-minute intervals. Atorvastatin, fluvastatin, and amlodipine mean Effective Concentrations (EC_50_) were noted [10,16]. The experiments were conducted four times.

### 2.8. Statins Effects on Calcium Concentration Response Curves (CCRCs)

Because statins relaxed the aortic strips and the KCL induced contractions, it is suggested that atorvastatin and fluvastatin may have effects via the calcium channel blockers. To sort out the underlying mechanisms, calcium concentration response curves (range: 1 × 10^−4^–256 × 10^−4^ Log calcium Molar solution) were constructed in the absence and presence of different concentrations of statins (where its relaxing effect was visible on 80 mM KCl induced contractions). A standard calcium channel blocker such as verapamil was used for comparison. Briefly describing the procedure, Krebs solution was used for aortic strip maintenance. For decalcification, the aortic strips were exposed to a series of wash with K-rich (Ca^++^ free) Krebs solution followed by exposure to K-rich Normal solution after a brief period of stabilization. Test concentrations of atorvastatin and fluvastatin were applied. An incubation period of 45–60 min was given. Then, CCRCs were constructed. Similarly, in the absence of atorvastatin and fluvastatin, control CCRCs were also constructed. Equally, the curves for verapamil were also drawn. Any possible right shift indicates the inhibition of voltage gated calcium channels [16,20,21,22,23].

### 2.9. Effects of Statins on Systolic Blood Pressure in Hypertensive Rats

Because statins relaxed the aortic strips, we transferred these effects using an in vivo model in Wistar rats. Hypertension was induced in rats using a Depomedrol IM injection. The rats were housed in the animal house of Khyber Medical University. Their blood pressure was checked regularly on alternate days. The rats were acclimatized for two weeks. Rats with systolic blood pressure greater than 140 mmHg (on 3 separate intervals) were considered hypertensive rats. The rats were divided into 4 groups containing 5 rats in each group. One group served as a negative control that only received the vehicle. The other group was treated with an equivalent dose of amlodipine 0.0714 mg/kg po that served as a positive control treatment. Statins in respective EC_50_ concentrations were administered to atorvastatin and fluvastatin treated groups via oral route. Their blood pressure was noted on respective Tmax. The experiments were conducted in quadruplicate [24].

### 2.10. Statistical Analysis

Statins test concentration effects on isolated rabbits’ aortic strips were graphically plotted versus test concentrations of statins as dose–response curves using the Graph Pad prism. Effects were expressed as a % of control maximum for NE and KCl induced contractions. For CCRCs control curves and curves in the presence of statins, log calcium molar concentration was plotted on the *X*-axis as an independent variable. Responses were plotted on the *Y*-axis as a dependent variable. Similarly, systolic blood pressure values were plotted in Graph Pad prism which was compared with amlodipine. The mean EC_50_ was calculated. One-way ANOVA and unpaired t-test were used for the determination of the significances of *p*-values < 0.05.

## 3. Results

The effects of atorvastatin on NE and KCl induced contractions in preparations are shown in Figure 1.

Similarly, the effects of fluvastatin and amlodipine on NE and KCl induced contractions are respectively shown in Figure 2 and Figure 3.

The derived relaxing effects of atorvastatin, fluvastatin, and amlodipine with their respective EC_50_ are shown in Table 1. It is noteworthy that statins relaxed both intact and denuded tissues. Maximum effects of atorvastatin, fluvastatin, and amlodipine as a % of their respective controls, i.e., % of NE (1 µM) and % of KCl (80 mM) induced contractions are also presented in Table 1.

It is noteworthy that the EC_50_ of fluvastatin on NE induced contractions is relatively less as compared to atorvastatin and amlodipine. This implies that fluvastatin is more potent compared to amlodipine and atorvastatin.

The effects of amlodipine as a standard calcium channel blocker in the presence of respective atorvastatin and fluvastatin are shown in Table 2, where EC_50_ on KCL induced contractions are less (a left shift) as compared to the EC_50_ of the statins tested alone. This further confirms that these statins have additional effects on the KCL induced contractions as well as on the NE (1 μM) induced contractions.

CCRCs in the absence and presence of statins are shown in Figure 4.

Derived EC_50_ for CCRCs are shown in Table 3. There is a right shift in EC_50_ indicating the involvement of statins for inhibition of voltage gated calcium channels.

It is evident from Table 3 that in the presence of atorvastatin (7.6 × 10^−7^ M, final bath solution), there is a right shift in EC_50_ −1.97 Log [Ca^++^] M versus control EC_50_, i.e., −2.81 Log [Ca^++^] M. Similarly, in the presence of fluvastatin test concentration (1.3 × 10^−6^ M), there is a right shift in EC_50_ −2.5 Log [Ca^++^] M versus its respective control EC_50_ i.e., −3.02 Log [Ca^++^] M.

The results of studying the effects of atorvastatin and fluvastatin on the systolic blood pressure of rats are shown in Figure 5. Fluvastatin showed a significant blood pressure lowering additional effect as compared to atorvastatin (*p* < 0.05) using amlodipine as standard.

## 4. Discussion

Clinicians treat most diseases with the administration of either two or more drugs. Such polypharmacy practices are very common while treating cardiovascular diseases such as hypertension, stroke, angina pectoris, and myocardial infarction. In patients who are hypertensive and hypercholesterolemic, a combination of calcium channel blockers and statins are frequently advised. Research conducted on cell lines states that statins up-regulate the L-type calcium channels and therefore have an additional effect on voltage gated calcium channels [11]. But this up-regulation requires time to develop. Our study examines the direct first dose effects of statins alone or if used in combination with a calcium channel blocker. We have recently reported that current statins have inhibitory effects on voltage gated calcium channels in intestinal preparations [10]. In light of these statements of recent research, the possibilities of drug–drug interactions cannot be ruled out. This study answers the possible vascular effects and subsequent drug interactions for a possible additional or synergistic effect.

The results imply that fluvastatin is more potent as compared to atorvastatin in denuded aortic strip preparations. This is evident from the amplitude of % of controlled response which remained up to 10% in NE (1 uM) induced contractions, while contractions remained with an amplitude corresponding to 36% of the control when treated with atorvastatin. Similarly, amplitudes for KCl (80 mM) induced contractions were relaxed up to 28% by fluvastatin as compared to 39% observed with atorvastatin in intact aortae. The relaxation produced by fluvastatin was even more potent than amlodipine, a standard calcium channel blocker. These effects are significant in intact and denuded tissues, which implies that atorvastatin and fluvastatin relax the aortic strips by more than one pathway. This is evident from the EC_50_ mentioned in Table 1, which shows that fluvastatin is more potent than amlodipine as far as its effects on NE induced contractions are concerned. Relaxing effects on KCL induced contractions are usually, but not necessarily, considered to follow the inhibition of voltage gated calcium channels (19). This was confirmed by plotting the CCRCs in the absence (control curve) and presence of tested statins (atorvastatin and fluvastatin). The right shift in the EC_50_ (Log Ca^++^ M) of CCRCs implies that statins have calcium channel blocking activity. A right shift in the EC_50_ of fluvastatin with a relatively lower EC_50_ value (−2.8 Log Ca^++^ M) in the presence of the test concentration (1.2 × 10^−7^ M) of fluvastatin implies that fluvastatin is more potent than atorvastatin. This right shift confirms that fluvastatin and atorvastatin have a vasorelaxant effect. This also confirms our previous report about statins that have a calcium channel blocking effect in isolated jejunal preparations. It is noteworthy that fluvastatin and atorvastatin relaxed the NE induced contractions and KCl induced contractions. This implies that these statins have a dual mode of actions through inhibition on receptor operated calcium channels and inhibition of voltage gated calcium channels in aortae. The literature reports that a relaxing effect on high 80 mM KCL induced contractions implies an inhibition of L-type calcium channels [16,23,24,25,26]. To detect the exact mechanisms for the involvement of inhibition of voltage gated L-type calcium channels, we constructed CCRCs where the right shift in the EC_50_ values confirms the calcium channel blocking activity of atorvastatin and fluvastatin in isolated aortae [16,27]. The relaxing effects on NE induced contractions imply involvement of the inhibition of calcium release from internal stores as the release of calcium, in a calcium free rich solution, is mediated through alpha 1 receptors. These findings point to a dual mode of action in the statins. These additional effects of test statins and standard calcium channel blocker amlodipine were examined in an in vivo study for analyzing its effects on systolic blood pressure. It is evident from Figure 5 that systolic blood pressure was significantly reduced (*p* < 0.05) when statins were added to amlodipine in hypertensive rats in their respective EC_50_ concentrations. This states that the additional effect of statins is dose-dependent and can be increased with an increase in the dose of these statins. The new aspect of test statins further increases the interest of pharmacotherapy in its inhibition of voltage gated calcium channels in addition to its inhibitory effects on NE induced contractions. This vasorelaxation effect can be of prime interest in stress induced vasoconstriction/hypertension associated with an increase in catecholamine surges during stress, which requires further study in a clinical setting. Thus, the dual action of statins makes it more interesting for studying its vasorelaxation phenomenon in a prospective clinical trial. The only concern is that the right dose is used to obtain these additional vasorelaxation effects. Currently, there are different meta-analyses which portray a lower effect on decreasing blood pressure as far as the statins current daily recommended hypocholesterolemic doses are concerned [15]. This could be because some statins are tested in doses approved for cholesterol lowering action. The current requirement is to study these statins on test concentrations where blood levels are simulated to these in vitro EC_50_ values, or to test these statins on a test concentration which relaxes the aortic strips to their maximum values. Here, we recommend a randomized clinical trial to analyze the new aspects of statins in prospective optimal dose(s) for a possible additional effect that may produce the inhibition of voltage gated calcium channels.

## 5. Conclusions

Atorvastatin and fluvastatin relax the aortic strip preparations predominantly through inhibition of voltage gated calcium channels in high molar KCL induced contractions. These statins also inhibit the effects of NE induced contractions. This study also confirms that atorvastatin and fluvastatin potentiate the blood pressure lowering effect of amlodipine in hypertensive rats.

## 6. Limitations

We did not study the effects of these statins on the release of calcium from internal stores.

## 7. Recommendations

Further studies are required to document the detailed mode of action on the internal release of calcium. The new aspect of statins can be analyzed in a clinical practice with a randomized clinical trial.

## Figures and Tables

**Figure 1 medicina-59-01023-f001:**
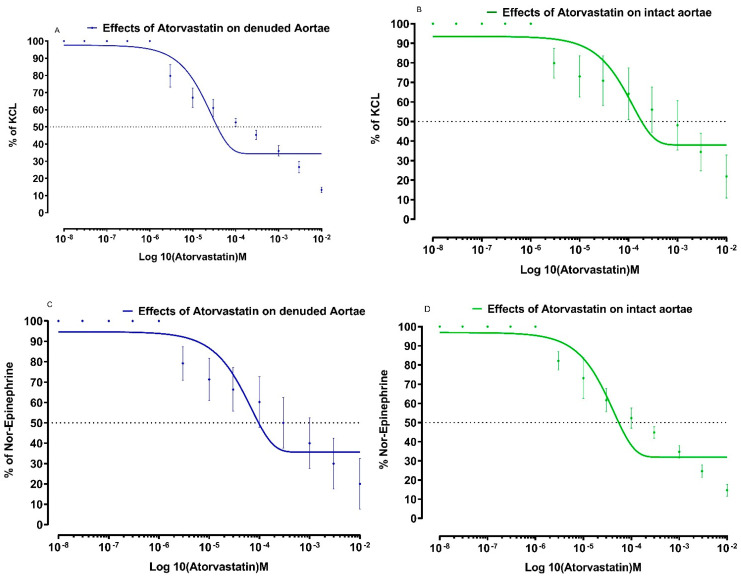
Effects of atorvastatin on isolated aortic strip preparations to show relaxant effects in intact and denuded tissues. (**A**) Effects of atorvastatin on KCL-induced contractions in isolated aortic strip preparations to show relaxant effects in denuded tissues. (**B**) Effects of atorvastatin on KCL-induced contractions in isolated aortic strip preparations to show relaxant effects in intact tissues. (**C**) Effects of atorvastatin on Nor-Epinephrine induced contractions in isolated aortic strip preparations to show relaxant effects in denuded tissues (all values are mean ± SD, *n* = 4). (**D**) Effects of atorvastatin on Nor-Epinephrine induced in isolated aortic strip preparations to show relaxant effects in intact tissues. All values are mean ± SD, *n* = 4.

**Figure 2 medicina-59-01023-f002:**
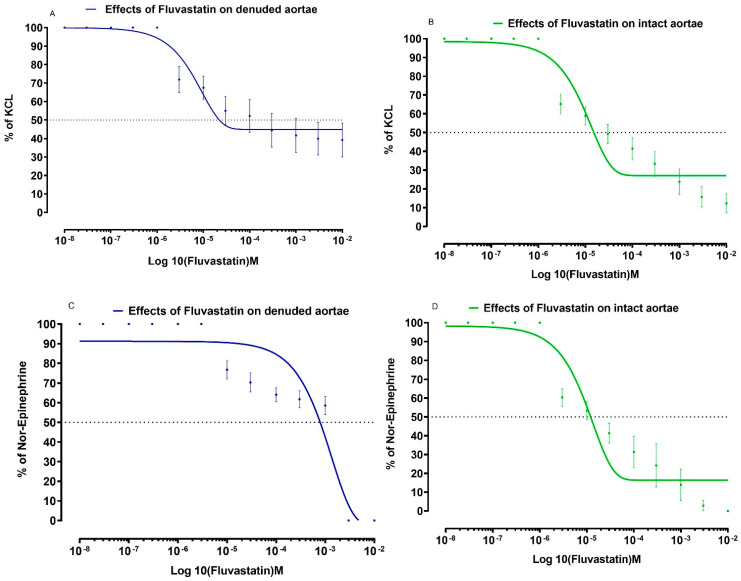
Effects of fluvastatin on isolated aortic strip preparations to show relaxant effects in intact and denuded tissues. (**A**) Effects of Fluvastatin on KCL-induced contractions in isolated aortic strip preparations to show relaxant effects in denuded tissues. (**B**) Effects of Fluvastatin on KCL-induced contractions in isolated aortic strip preparations to show relaxant effects in intact tissues. (**C**) Effects of Fluvastatin on Nor-Epinephrine induced contractions in isolated aortic strip preparations to show relaxant effects in denuded tissues. (**D**) Effects of Fluvastatin on Nor-Epinephrine induced in isolated aortic strip preparations to show relaxant effects in intact tissues. All values are mean ± SD, *n* = 4.

**Figure 3 medicina-59-01023-f003:**
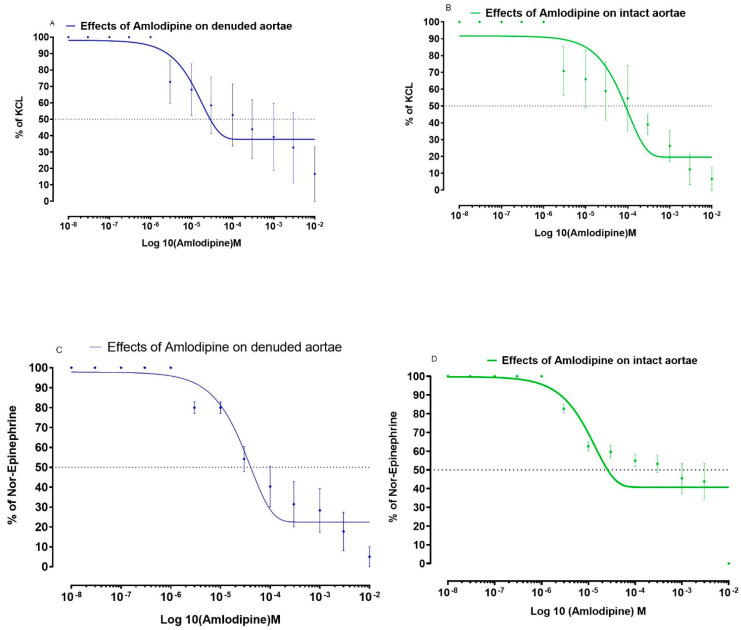
Effects of amlodipine on isolated aortic strip preparations to show relaxant effects in intact and denuded tissues. (**A**) Effects of amlodipine on KCL-induced contractions in isolated aortic strip preparations to show relaxant effects in denuded tissues. (**B**) Effects of amlodipine on KCL-induced contractions in isolated aortic strip preparations to show relaxant effects in intact tissues. (**C**) Effects of amlodipine on Nor-Epinephrine induced contractions in isolated aortic strip preparations to show relaxant effects in denuded tissues. (**D**) Effects of amlodipine on Nor-Epinephrine induced in isolated aortic strip preparations to show relaxant effects in intact tissues. All values are mean ± SD, *n* = 4.

**Figure 4 medicina-59-01023-f004:**
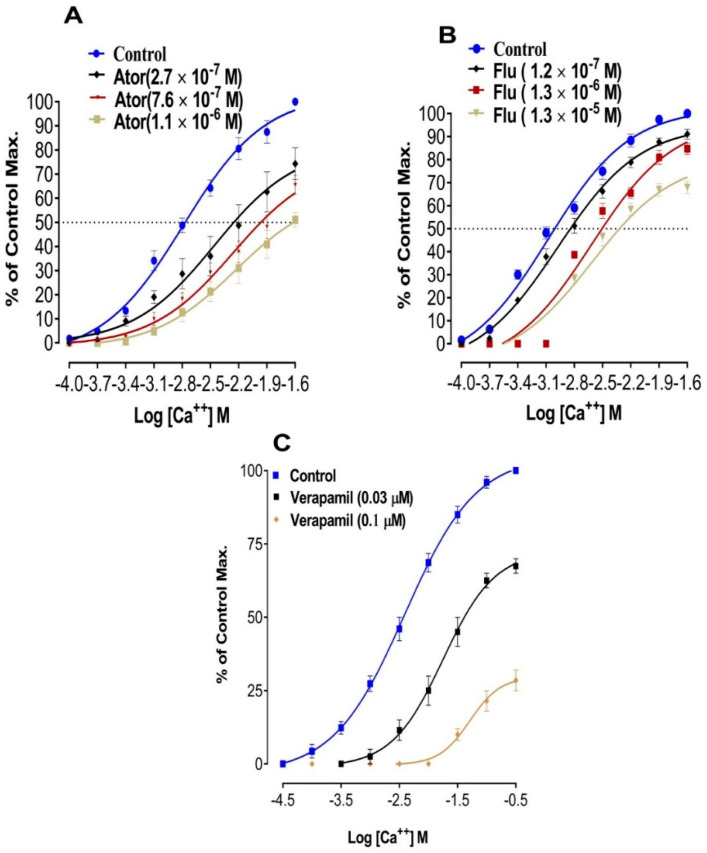
CCRCs in the absence and presence of statins. (**A**) To show construction of CCRCs in the absence and presence of different concentrations of atorvastatin in isolated aortic strip preparations (all values are mean ± SD, *n* = 4). (**B**) To show construction of CCRCs in the absence and presence of different concentrations of fluvastatin in isolated aortic strip preparations (all values are mean ± SD, *n* = 4). (**C**) To show construction of CCRCs in the absence and presence of different concentrations of verapamil in isolated aortic strip preparations (all values are mean ± SD, *n* = 4).

**Figure 5 medicina-59-01023-f005:**
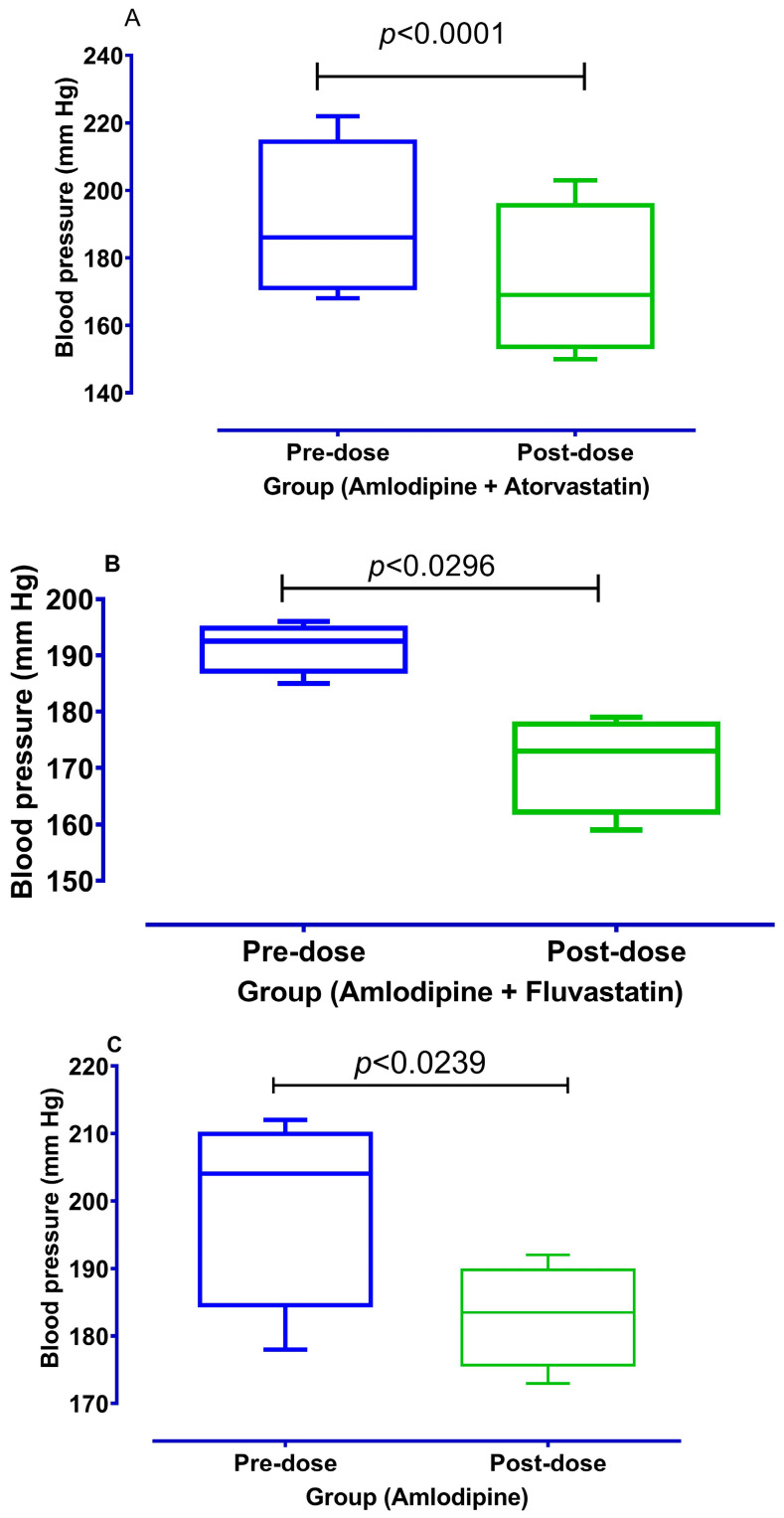
To show additional effect of atorvastatin and fluvastatin on the blood pressure lowering effect of amlodipine (all values are mean ± SD, *n* = 4) using Paired *t*-test at 95% Confidence Interval. (**A**) To show additional effect of atorvastatin with amlodipine on systolic Blood pressure. (**B**) To show additional effect with fluvastatin with amlodipine on systolic Blood pressure. (**C**) To show effect of amlodipine on systolic Blood pressure.

**Table 1 medicina-59-01023-t001:** To show relaxing effects of atorvastatin, fluvastatin, and amlodipine with their respective EC_50_ values (all values are mean ± SD, *n* = 4).

Drugs	Aortae Status	% of KCL (Control Max)	% of NE (Control Max)	EC_50_ ± SD KCL Induced (Molar)	EC_50_ ± SD NE Induced (Molar)
Atorvastatin	Denuded	34.4%	36%	3.64 × 10^−5^ ± 2.1	9.76 × 10^−5^ ± 3.5
	Intact	39%	33%	1.85 × 10^−6^ ± 4.1	5.3 × 10^−5^ ± 3.0
Fluvastatin	Denuded	45.4%	10%	4.78 × 10^−5^ ± 2.25	8.31 × 10^−4^ ± 1.49
	Intact	28%	17.3%	1.46 × 10^−5^ ± 1.73	1.23 × 10^−5^ ± 1.35
Amlodipine	Denuded	39.1%	22.7%	3.29 × 10^−5^ ± 1.69	3.83 × 10^−5^ ± 1.52
	Intact	40%	40.6%	2.3 × 10^−5^ ± 1.11	2.52 × 10^−5^ ± 1.81

**Table 2 medicina-59-01023-t002:** To show relaxing effects of atorvastatin, fluvastatin, and amlodipine with their respective EC_50_ values (all values are mean ± SD, *n* = 4).

Drugs	Aortae Status	% of KCL (Control Max)	% of NE (Control Max)	EC_50_ ± SD KCL Induced (Molar)	EC_50_ ± SD NE Induced (Molar)
Amlodipine + Atorvastatin	Denuded	13%	12.8%	1.30 × 10^−5^ ± 2.1	4.38 × 10^−6^ ± 1.5
	Intact	16.2%	15.2%	2.52 × 10^−5^ ± 0.01	1.26 × 10^−5^ ± 0.02
Amlodipine + Fluvastatin	Denuded	10%	10.6%	6.50 × 10^−6^ ± 2.25	2.01 × 10^−6^ ± 1.49
	Intact	12%	13%	3..87 × 10^−6^ ±1.73	3.1 × 10^−6^ ± 1.35

**Table 3 medicina-59-01023-t003:** To show right shift EC_50_ of atorvastatin, fluvastatin, and verapamil on calcium concentration response curves (all values are mean ± SD, *n* = 4).

Statins	CCRCs Specifications	EC_50_ Log [Ca^++^] M
Atorvastatin	Control	−2.81
	Test Concentration (2.7 × 10^−7^ M)	−2.27 **
	Test Concentration (7.6 × 10^−7^ M)	−1.97 **
Fluvastatin	Control	−3.02
	Test Concentration (1.2 × 10^−7^ M)	−2.83 **
	Test concentration (1.3 × 10^−6^ M)	−2.50 **
Verapamil	Control	−2.42
	Test Concentration (0.03 µM)	−1.41 **
	Test concentration (0.1 µM)	−0.69 **

** = Highly significant versus verapamil (standard) using one-way ANOVA, *p* < 0.05.

## Data Availability

The datasets (Power Lab data) during and/or analyzed during the current study are available from the corresponding author on reasonable request.

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
