# Peer review of "Atorvastatin and Fluvastatin Potentiate Blood Pressure Lowering Effect of Amlodipine through Vasorelaxant Phenomenon"

_medicina, 2023, doi:10.3390/medicina59061023_

Round 1

Reviewer 1 Report

Dear Editor, Dear Authors,

Thank you very much for this exciting research question and the prepared manuscript.

Nevertheless, there are aspects that, in my opinion, should be worked on before publication. I would shorten the abstract a little more and work out the essential points. Likewise, the question would be whether the title could include the animal experiment in the title.

The introduction and the methods section are clearly structured and provide a good introduction to the topic. In the results section, I would appreciate it if the graphs could be made with dot representation for each individual measuring point. This would provide greater clarity and transparency. This is especially true for Graph 5. I would also welcome significance in Graph 5.

Author Response

Dear reviewer the response to yours worthy comments is hereby attached please.

Reviewer 2 Report

The manuscript describes some important concepts, which I have found quite interesting. However, I would like to suggest some some modifications to improve the quality of the review.

The abstract needs to be divided in clear sections like background, methods, results discussions etc to give an easy overview to the readers. 

The introduction needs to be elaborated. More recently published articles should be reviewed to improve the quality of the manuscript.

The authors should compare this study with previous published ones so as to give a clear indication how this article holds novelty.

The objective of the study should be made more clear.

The conclusions section is not up to the mark and needs to be improved

Moderate editing of English language

Author Response

(The authors gave the same response as above.)

Reviewer 3 Report

medicina-2384976, Atorvastatin and fluvastatin potentiate blood pressure lowering effect of amlodipine through vasorelaxant phenomenon by Niaz Ali et al. The authors aimed to study the effects of atorvastatin and  fluvastatin on blood vessels for possible vasorelaxant effect and subsequent effect on systolic  blood pressure. The study also focuses on drug-drug interaction of the atorvastatin and fluvastatin, and amlodipine, when used in combination. 

Comments:

Abstract:

- Page 1: The abbreviations NE and CCRCs should be fully explained at their first mention in the manuscript.

Introduction:

- Page 1, line 43: Could the authors add the abbreviation of “cardiovascular disease”.

- Page 1, line 44: “In 2017, 17.8 million deaths occurred due to CVDs”. Could the authors update the reported number of deaths to a more recent year and provide a reference for the new data.

Page 2: Could the authors provide full explanation of the abbreviations WHO and HMG-CoA.

- Page 2, lines 51-52: “WHO approved drugs for CVDs treatment are Beta blockers….Statins and antiplatelet drugs”. The reviewer suggests inserting the phrase “such as” to the sentence “WHO approved drugs for CVDs treatment such as Beta blockers”.

- Page 2, lines 66-67: The reviewer suggests adding the drug-drug interaction of the atorvastatin and fluvastatin, and amlodipine as part of the aim.

Materials and methods:

- Page 2, line 81: Could the authors add the abbreviation of “carboxy methyl cellulose”.

- Page 2, line 84: “experience”. Wrong spelling.

- Page 3, lines 100-103: “Three types of Kreb’s solutions were used: 1- Krebs’s normal…were used on the same day of experiments”. The reviewer suggests providing a clearer description of the physiological solutions used in the manuscript.

- Page 3, line 111: “Ach”. Could the authors spell the abbreviation. 

Page 3, 117: “NE”. Spell the abbreviation. 

- Page 4, line 151: “I/M injection” should be written IM, please correct accordingly.

Results: 

- The reviewer suggests changing the titles of figures 1,2 and 3 to accurately describe the effect of the drugs on N.E and KCl induced contractions.  

Table 3: What does the asterisk mark signify in this context, and which statistical method was used to analyze the data? Could the authors clarify in the table footnote. 

Discussion:

- Page 9, lines 239-241: “10% of N.E (1uM) induced… 39% observed with atorvastatin”. Could the authors check if the numbers reported are accurate with that in the table. Could you also clarify if the comparison being made is between nuded and denuded?

- Page 9, lines 257-260: “It is noteworthy fluvastatin and … calcium channels in aortae”. Could the authors provide additional information regarding the proposed dual action and support it with relevant references, if available.

- Page 9, lines 273-275: “As till now, there are different … hypocholesterolemic doses are concerned”. What the authors are referring to? Could you provide a brief explanation?

Conclusion:

The reviewer suggests rewriting the conclusion to emphasize the key findings of the study and highlight the novelty of the research.

Moderate editing of English language

Author Response

(The authors gave the same response as above.)

Round 2

Reviewer 1 Report

Dear editors, dear authors, thank you very much for the revised version. Furthermore, aspects of the presentation of results are only addressed to a limited extent. Otherwise, the publication has clearly become accustomed to quality.

Author Response

Dear Reviewer once again thanks for yours input.Response to yours worthy comments is hereby attached please.

Reviewer 2 Report

The revised version of the manuscript can be recommended for publication

Author Response

(The authors gave the same response as above.)

Reviewer 3 Report

I would like to thank the authors for revising this manuscript and addressing the reviewer comments. While the authors have done a thorough job of addressing the previous comments, I have few additional comments for them to consider:

Materials and methods:

- Page 2, line 81: Could the authors add the abbreviations of “Acetylcholine, Nor-epinephrine, and potassium chloride”.

- Page 3, The reviewer suggested providing a clearer description of the physiological solutions used in the manuscript. Although the authors indicated their intention to include the description as advised, it has not been incorporated into the manuscript yet.

Minor editing of English language required

Author Response

(The authors gave the same response as above.)
